# Circular Economy on Construction and Demolition Waste: A Literature Review on Material Recovery and Production

**DOI:** 10.3390/ma13132970

**Published:** 2020-07-03

**Authors:** Clarence P. Ginga, Jason Maximino C. Ongpeng, Ma. Klarissa M. Daly

**Affiliations:** Department of Civil Engineering, De La Salle University Manila, 2401 Taft Avenue, Malate Manila 0922, Philippines; jason.ongpeng@dlsu.edu.ph (J.M.C.O.); ma.klarissa.martinez@dlsu.edu.ph (M.K.M.D.)

**Keywords:** circular economy, construction, demolition, waste, recycling

## Abstract

Construction and demolition waste (CDW) accounts for at least 30% of the total solid waste produced around the world. At around 924 million tons in the European Union in 2016 and 2.36 billion tons in China in 2018, the amount is expected to increase over the next few years. Dumping these wastes in sanitary landfills has always been the traditional approach to waste management but this will not be feasible in the years to come. To significantly reduce or eliminate the amount of CDW being dumped, circular economy is a possible solution to the increasing amounts of CDW. Circular economy is an economic system based on business models which replaces the end-of-life concept with reducing, reusing, recycling, and recovering materials. This paper discusses circular economy (CE) frameworks—specifically material recovery and production highlighting the reuse and recycling of CDW and reprocessing into new construction applications. Likewise, a literature review into recent studies of reuse and recycling of CDW and its feasibility is also discussed to possibly prove the effectivity of CE in reducing CDW. Findings such as effectivity of recycling CDW into new construction applications and its limitations in effective usage are discussed and research gaps such as reuse of construction materials are also undertaken. CE and recycling were also found to be emerging topics. Observed trends in published articles as well as the use of latent Dirichlet allocation in creating topic models have shown a rising awareness and increasing research in CE which focuses on recycling and reusing CDW.

## 1. Introduction

The construction industry and demolition process after the expiration of the life-cycle of a building or structure is one that produces a considerable amount of waste. In the European Union, approximately 930 million tons in 2016 [1] and 2.36 billion tons in China in 2018 [2]. The mentioned industry is responsible for the production of considerable amounts of waste and the increasing volume has become unbearable for the environment, economic, and social viewpoints [3]. Construction and demolition waste or CDW, is a kind of solid waste that arises from construction sites and in total or partial demolition of buildings and infrastructures. Construction waste is due to excessively ordered supplies or mishandling of materials by unskilled laborers. Demolition is the removal of outmoded and unusable structures to replace with new ones. In some instances, CDW can also be generated following a natural disaster [4]. Studies in CDW have long been done since the late 20th century and problems associated with building waste that usually consists of sizable units of debris in large volumes [5]. The lack of knowledge on the composition and characteristics by many who manage CDW results in the dumping of huge quantities of potentially recyclable materials which could be an alternative to their natural counterparts [3]. CDW is a major challenge for the construction and demolition industry due to the increasing volume of waste produced and its associated environmental impacts. CDW is the largest waste worldwide at around 30 to 40%: 36% in the European Union, and close to 67% in the United States [6]. In countries with numerous research and review articles on construction and demolition waste like China, Figure 1 shows the detailed flow of CDW that is processed for recycling in Beijing. The recycling rate in Beijing is only 3%; 70% to 80% of CDW is discarded to landfills and about 10% CDW is burned directly or used as fuel [2].

Circular economy or CE is an economic system that is based on business models which replace the “end-of-life” concept—a stage of any product that does not receive continuing support, either because existing processes are terminated or it is at the end of its useful life —with reducing or alternatively reusing, recycling, and recovering materials in the production/distribution and consumption processes [7]. This infers less overall waste produced and discarded from both manufacturing and raw materials processing [8]. The concept of CE implies a mindset change that considers waste as a potentially useful resource and not as a problem to manage and dispose [9]. CE is considered a solution as it would reduce environmental impacts while contributing to economic growth. As early as 1966, awareness of circular economy was made by Kenneth Boulding, the economics of the coming spaceship Earth is often cited as the first expression of the circular economy although Boulding does not use that term. The emergence of an economy in loops or circular economy was introduced by Walter Stahel and Genevieve Reday in their 1976 research report to the European Commission, the potential for substituting manpower for energy which analyzed manufacturing of cars and construction of buildings on a micro and macroeconomic basis. The CE concept has gained academic, government, and organizational recognition. At a global level, Germany, Japan, China, and Europe are recognized for having developed legislation to the implementation of CE principles [6]. The CE system in the construction and demolition industry has five influential stages: preconstruction, construction and building renovation, collection and distribution, end-of-life, and material recovery and production [6]. This paper studies the material recovery and production and the current research and breakthroughs over the past years.

## 2. Methodology

A comprehensive and modern method of literature review was done to address the research objectives previously stated. Relevant literature was found using a keyword-based search from electronic databases such as https://www.scopus.com and was explored by typing the keywords "circular economy" AND "construction" OR "demolition" AND "waste" OR "recycling". From the results, the search is narrowed down from documents published in 2016 to 2020 so as to review recent studies done in the field to avoid reviewing outmoded studies. A total of 360 documents were found. From the 360 documents, by manually reading the abstracts and parts of the entire article, and considering the following criteria: Studies that assess and discuss the use of recovered materials in the manufacturing of new construction materials from a CE perspective.Studies that assess reuse, recycling, and recovery of CDW from an environmental perspective.Studies that assess and discuss the effectivity of recovered, reused, or recycled CDW from a mechanical/performance perspective.Studies that discuss effective CE framework in CDW material recovery and production.
Papers that qualified to at least one of the mentioned criteria, were not redundant to other articles, and were critiqued as papers focusing only on CE, CDW, and reuse and recycling of CDW in new construction applications were included, which gave a total of 34 papers. Early studies and published papers were also included in this review article, thus showing that studies in CE in CDW were made as early as the 20th century. A review of the research articles was conducted in order to review existing CE frameworks focusing on material recovery and processing and to assess the present reuse and recycling strategies of CDW and its competitiveness with its virgin counterparts. 

Utilizing Matlab text data analytics, a software that makes data science easy with tools to access and preprocess data, build machine learning and predictive models, and deploy models to enterprise IT systems, the abstracts were preprocessed applying a Latent Dirichlet Allocation model or LDA, which is a particularly common method for fitting a topic model and is a kind of algorithm that is a three-level hierarchical Bayesian-modelling process which groups a set of items into topics defined by words or terms [10] to discover the underlying topics among the articles. Using LDA, four primary topics were obtained and are as follows:Topic 1: demolition, waste, material, recyclingTopic 2: circular economy, construction, environment, transitionTopic 3: research, potential, reuse, buildingTopic 4: aggregate, concrete, strength, mechanical
The topics from the 34 included papers focus on the four primary topics as shown above and in Figure 2. Topic mixtures and probabilities are shown in Figure 3. Topic 1 primarily focuses on waste produced in the construction and demolition industry and the recycling of such materials to new applications. Topic 2 focuses on the circular economy; the transition into and the environmental impacts of CE. Topic 3 focuses on the research on the potential of reuse in building materials. Lastly, topic four focuses on the research of mechanical properties or performance of reused or recycled materials. Figure 4 demonstrates the trend and number of papers published from 2016 to 2020 on the topic showing the increasing awareness on CDW among researchers.

## 3. CE Framework on Material Recovery and Production

The CE framework when discussed in whole is broad, thus, this paper has given focus to the material recovery and production which deals with the reuse and recycling of CDW in new construction applications. The potential of material reuse and recycling to reduce environmental impacts associated with construction and demolition has already gained recognition among policy makers [11]. Despite the gained recognition and market potential of CDW being reintroduced into new construction applications, it is still hindered by barriers such as logistics (41%), cost (29%), regulations (12%), and others (6%) [12]. There is also a negative attitude towards reused and recycled products perceived by many as environmentally friendly but of lower quality [9]. An effective framework into the material recovery and production would significantly reduce if not completely eliminate the barriers to CE.

In a broader perspective, Figure 5 demonstrates the basic process of how CDW is produced, processed, and dumped into landfills, or is reused and recycled. Excessively ordered or materials handled by inexperienced labor leads to the production of CDW in construction projects while materials left after demolition which are not designed for deconstruction are likewise turned to CDW. Landfill disposal and incineration are the usual endpoint of this CDW. The goal of a CE is to reduce if not eliminate CDW being dumped into landfills and incinerated, but also focuses on expanding the scale and quality of CDW recycling and reuse and its potential to construct new buildings [12].

An effective framework in the CE requires three strategies [13]:Narrowing resource loops—use of less material input for production in order to have less waste output at the end of life.Slowing loops—this means the lengthening of the use phase of materials.Closing resource loops—this can also be equal to the process of recycling of materials.

In material recovery and production, specifically in the reuse and recycling of materials, closing resource loops is the main strategy employed for an effective framework in the reuse and recycling of CDW. The recirculation of recovered resources in the life cycle allows the use in new construction applications, avoiding the use of virgin raw materials [6]. Material reuse is the practice of using applicable building materials again while recycling requires the breaking down of used items to make new materials and objects [2]. In Figure 6 as shown, this visually describes the CE framework for material recovery and production [6]. 

In this framework which could be related in Figure 5 with the application of the study by Baldasarre et al., materials to be reused are either directly reused without additional processes, by applying reprocessing methods such as repairing, refurbishing, and remanufacturing; other materials can also be reused. Depending on the material quality standards, recycling can either be closed, semi-closed, or open-looped recycling [6]. CDW material recovery and production in the CE should be an integral part of the economy; reuse and recycling CDW could save landfill, save energy and reduce greenhouse gas emissions, and achieve environmental sustainability [14]. Following the framework effectively would lead to extending of product/material value, provide long life to the material, and extend the resource value of CDW [15].

## 4. CDW Material Reuse and Recycling in CE

CDW reuse and recycling has already been studied since the late 20th century with papers published which examined the reuse of materials from construction and demolition [16]. While frameworks that promote reuse and recycling of CDW can be seen in the early 2000’s in the paper published which acknowledges the amount of CDW that is sent to landfills and severely damages the environment [17,18]. The application of the framework of CDW material recovery and production specifically in the reuse and recycling of CDW can be seen as an emerging topic due to the number of experimentations and construction applications done. From 2016 to present, numerous papers have been published in the study of reuse and replacing materials with CDW alone. This does not account for other papers published from the late 20^th^ century discussing CE and reutilization of CDW.

Material reuse either by direct use or by repair/refurbishment as published that promotes CE by ensuring CDW is reused within the construction industry [19]. This approach minimizes leakages that hinder sustainable reusing of CDW [20]. Interventions for promoting reuse are already established [21]:Adaptive reuse—is a method that reuses whole or part of a structure that is redundant.Deconstruction—is the careful dismantling to maximize the recovery of components to be reused.Design for deconstruction (DfD)—is a designing method that closes construction component loops.Design for reuse (DfR)—incorporates the use of reclaimed components in the design of new structures.

For better understanding of the enumerated interventions in promoting reuse specifically DfD which could also be related to the other three interventions mentioned, Figure 7 shows a simplified life cycle of a building comparing the conventional design, construction, and disposal of demolished structures as shown in grey. The conventional method involves a linear approach that does not minimize or lessen the production of CDW dumped in landfills or incinerated. This is compared to the method incorporating DfD, which promotes the design of building materials that are capable of being deconstructed, remanufactured, and reused in new construction applications. This circular approach to a conventional method promotes minimized production of CDW that burdens the environment and threatens the sustainability of the construction and demolition industry.

Material recycling as discussed by papers published that promote closed-loop or recycling that involves breaking down or reprocessing of CDW into new materials [4,22]. Figure 8 and Figure 9 show the offsite and onsite framework that can be adopted in recycling CDW, respectively [14]. Recycling should be sorted and classified to either inert or non-inert CDW. Inert CDW refers to CDW that is neither chemically nor biologically reactive and will not decompose [23]. Adopting the framework for recycling CDW is a form of extending resource value which is the collection or sourcing of wasted materials and resources to turn these into new form of value [15]. For a deeper understanding of the reuse and recycling of CDW, Figure 10 shows a single-case study design with three sub-units: concrete, wood, and glass [11]. Figure 10 also illustrates how three commonly used materials are obtained, stored, processed, and reused and recycled. The framework provides an in-depth description that can also be adopted in future study, evaluation, and creation of frameworks. The narrative produces an example with reduced complexity that can play an important role in the development in the reuse and recycling of CDW.

Table 1 shows the differentiation of the latest experimentation, research, and studies conducted from 2016 to present and the observation on the effectiveness of the reuse and recycling of CDW in CE. Existing review-based studies have targeted the applications of recycled aggregate in concrete production, especially investigation of properties of recycled aggregate concrete containing recycled aggregate [24].

Among the applications of CE in CDW recovery and production, reuse of CDW is the optimal management measure due to it having the lowest adverse impacts [2]. However, lack of knowledge and an underdeveloped market reduces the capability of CDW reduction in a CE system [2].

Apart from the literature reviewed in Table 1, Table 2 shows a similar review on the recycling of CDW and its applications into new construction [43]. The review covered the years from as early as 1993 to as recent as 2016. Most applications of CDW lean toward the recycling of materials. It can also be inferred that the recycling and reprocessing of concrete aggregates was common place at more than 95% of the reviews done by Silva et al., and this paper. The recycling and reprocessing of aggregates have mixed results, having attained the required mechanical properties and conditionally achieving the required mechanical properties. RA are highly porous compared to natural aggregates [38], this means that water absorption is at least 6 times higher and 19% less dense as RA’s virgin counterpart. It is feasible to use RA if limited to up to 40% volume of the mix design [35] or replacement ratios of under 50% [33]. The addition of cement to compensate the strength loss due to use of recycled materials may also be done as an option [38]. A notable study was done wherein through patented technology of separating the hardened cement mortar and the coarse aggregate in the concrete rubble produced waste-free recycling in which optimal parameters were developed so as to recover high quality RA that improves concrete’s strength by 10% without significant decrease in other properties as well as the recovery of fine material (recycled concrete mortar) that can be used for manufacturing autoclaved materials in the amount of up to 20% of lime and sand mass was also obtained [42].

The use of recycled aggregates in a circular economy has been commonplace since aggregates are the most versatile material in construction that can be replaced as discussed earlier in this paper. Several construction applications around the world have adopted the use of recycled CDW as aggregates. In a compilation of case studies around the world as shown in Table 2, the technical feasibility of using recycled aggregates was explored in a wide range of new construction applications. Among these applications were unbound, hydraulically bound, bitumen bound, rigid pavement construction, and concrete applications. Table 2 shows the summary of the construction applications where CDW aggregates were used and the corresponding results are indicated.

Based on the compilation of study cases in Table 2, successful application of new materials, especially CDW as aggregates in new construction applications, is the best way to raise awareness into applying the circular economy in the construction industry. With the success of using CDW in the new construction applications, not only engineers but other professionals may be able to see the underlying benefits economically and in an environmental standpoint.

Looking into the studies made in reusing CDW, in an environmental standpoint to achieve the most beneficial positive effect in reuse of materials, rates of 70% for energy use and carbon emission are needed and 40% for water use. Transportation is another factor since transport distances of greater than 3000 km for reuse negate the beneficial effect [19]. However, findings in reuse cannot be considered due to the lack of supporting studies and research done in this conclusion. Of the 360 documents that were found as described in Section 2, the majority of which focus on recycling of CDW in new construction applications. However, it is good to note that there are studies in the past years that investigate the reusability of materials. 

Table 3 summarizes the different reuse potential rates or embodied carbon (EC) reuse efficiency of a range of construction components from 2000 to 2014. Potential rates are the measure of the ability of components to retain its functionality after the end of its primary life [21]. There are also other studies that evaluate the reusability of materials as discussed in earlier portions of the paper but Table 3 presents an in-depth analysis of various construction components that could help future research and experiments into reuse of CDW in new construction applications.

## 5. Research Gaps and Emerging Topics

The recycling and reutilization of CDW in new construction applications has been studied as early as the late 20^th^ century. However, research gaps such as applications of reuse into new construction applications are yet to be explored. Further, studies into reuse are limited or fewer in number, nevertheless studies in recycling outnumber greatly the studies into reuse as can be inferred in Figure 3 due to it having the least probability among the topics included in the chosen references. Despite a number of initiatives to unlock the potentials of reuse, lack of quantitative information restricts the advantages to be gained [21]. Most research and studies made in the past 30 years have been in the effective recycling of CDW in new construction applications. 

CE and recycling are emerging topics as can be seen by the trend of published studies and papers in Figure 4 that has seen a lot of potential in not only reducing CDW but also gives the opportunity for the construction and demolition industry to maintain a sustainable development. Also, as seen in Figure 3, the included papers show a high probability of having topics relating to CE and recycling. Various frameworks have already been established and CE has already achieved international recognition [6]. In recycling, a considerable number of studies has already been made in the reprocessing of CDW in new construction applications. 

## 6. Conclusions and Recommendation

Current frameworks in CE, CDW, and material recovery and production focusing on reuse and recycling show a consistent drive into promoting a CE in the construction and demolition industry to minimize if not totally eliminate the high production of CDW that threatens the environment and raises sustainability issues. These frameworks provide guidelines to future research and urge development into a more effective CE where instead of a linear approach to the design, construction, demolition, and disposal of CDW that produces the vast amounts of CDW, a circular model or approach that allows materials to be reprocessed or remanufactured, prolonging the life-cycle of the material therefore alleviating the rising number of CDW disposed. CE on material recovery and processing specifically on recycling CDW into new construction applications is seen as a feasible approach to be done due to the different applications of CDW when reprocessed and remanufactured into new construction materials. Based on the various research, tests, and results found in recycling materials, construction materials with recycled components present almost the same physical and mechanical properties as those of their virgin counterparts. In cases where the mechanical properties are slightly lower than their natural counterparts, amounts of other materials are used to compensate for the slight decrease which is negligible compared to the environmental and sustainability benefits of recycling CDW. 

Since the purpose of this study is to provide guidance to future work into CE, CDW, and material recovery and production focusing on recycling and reuse, further work and studies should be focused on measures for effective reuse of construction materials. The amount of research done on recycling greatly outnumbers the research done on reuse. In terms of material recycling and reprocessing, experimentation on effective proportioning of recycled materials, natural materials, and other materials should be studied to maximize the use of recycled CDW. Further studies into methods that promote waste-free recycling are also suggested since this is fully in line with the ideology of CE and promotes sustainable development and environmental preservation. In terms of experimentation and study on the physical and mechanical properties of recycled materials; currently, around 40% or lower amounts of recycled CDW are feasible in new construction applications. Additives can also be studied so that the use of 41% or more recycled CDW can be optimized. In cases where 100% replacement were found to be viable, further studies should be done to reduce any risk that could arise in the use of CDW instead of their natural counterparts especially in structural/load-bearing use. Nonstructural applications of 100% replacement of recycled materials is viable. 

## Figures and Tables

**Figure 1 materials-13-02970-f001:**
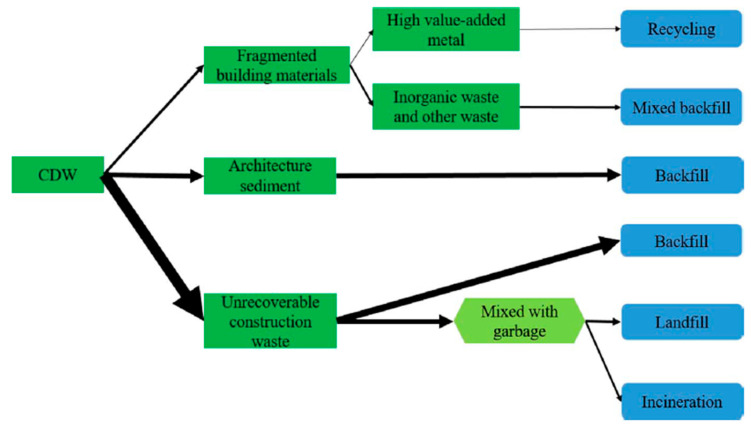
Construction and demolition waste (CDW) flow in Beijing [2].

**Figure 2 materials-13-02970-f002:**
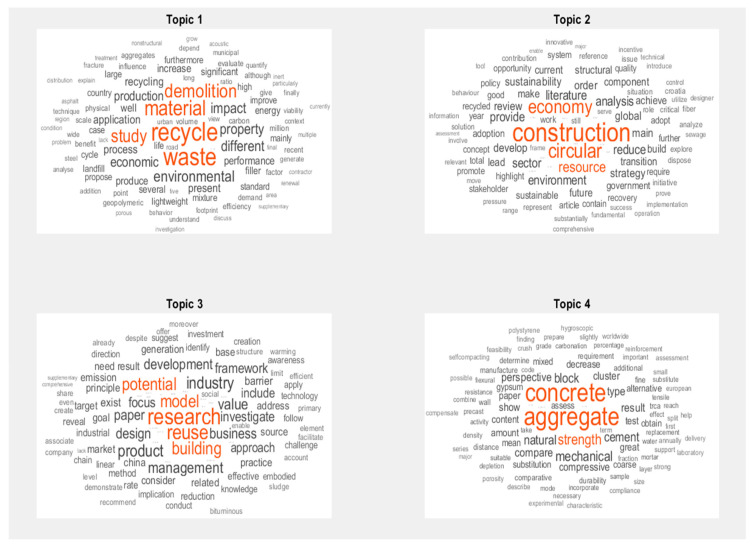
Primary topics found in the 35 included papers using Matlab text data analytics.

**Figure 3 materials-13-02970-f003:**
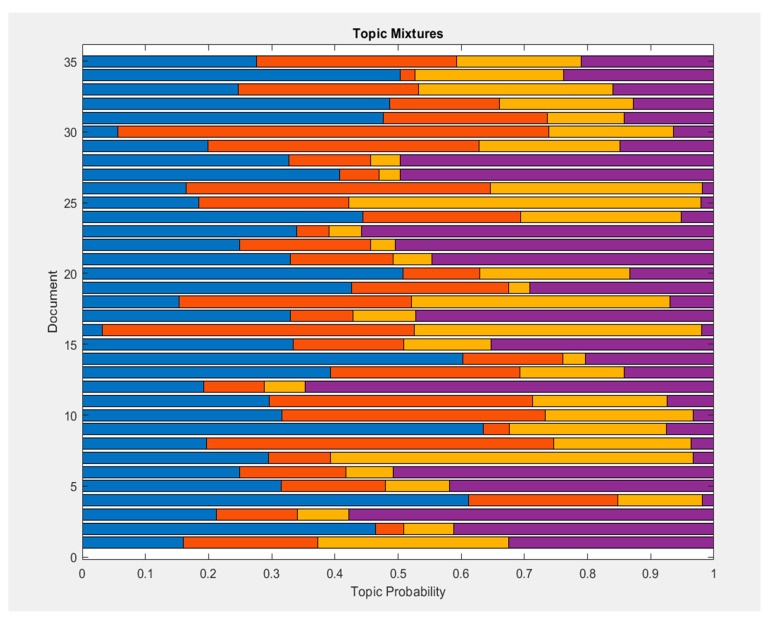
Topic mixtures with probabilities found in the 35 included papers using a latent dirichlet allocation model (LDA). Blue = Topic 1; Orange = Topic 2; Yellow = Topic 3; Violet = Topic 4.

**Figure 4 materials-13-02970-f004:**
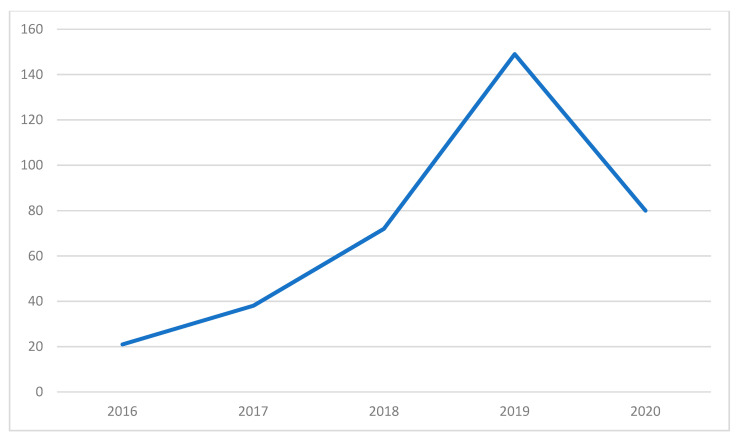
Number of papers published per year on the topic.

**Figure 5 materials-13-02970-f005:**
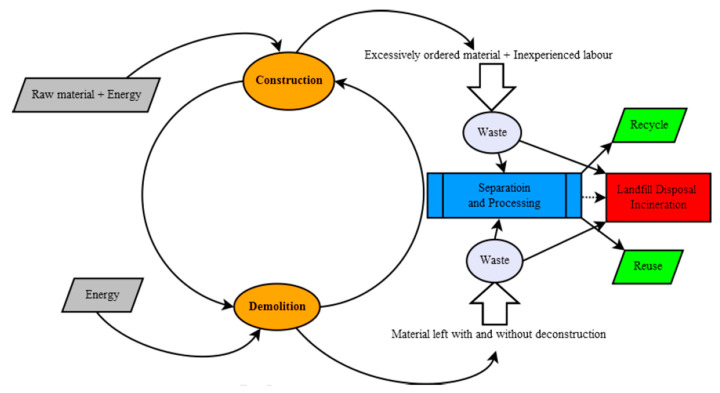
Schematic diagram of construction and demolition materials from beginning to end-use/disposal [4].

**Figure 6 materials-13-02970-f006:**
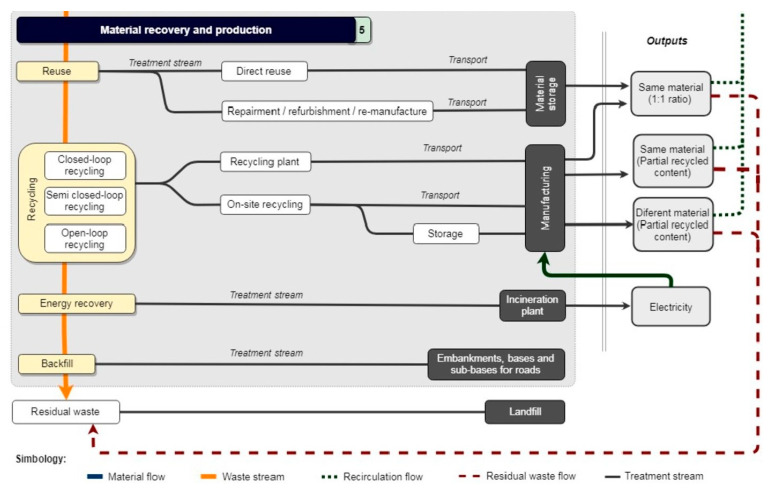
CE framework for material recovery and production [6].

**Figure 7 materials-13-02970-f007:**
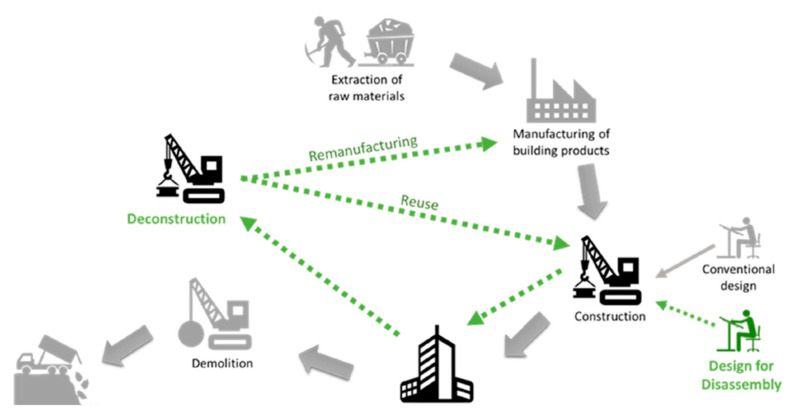
Building life cycle with conventional method and design for deconstruction (DfD) [19].

**Figure 8 materials-13-02970-f008:**
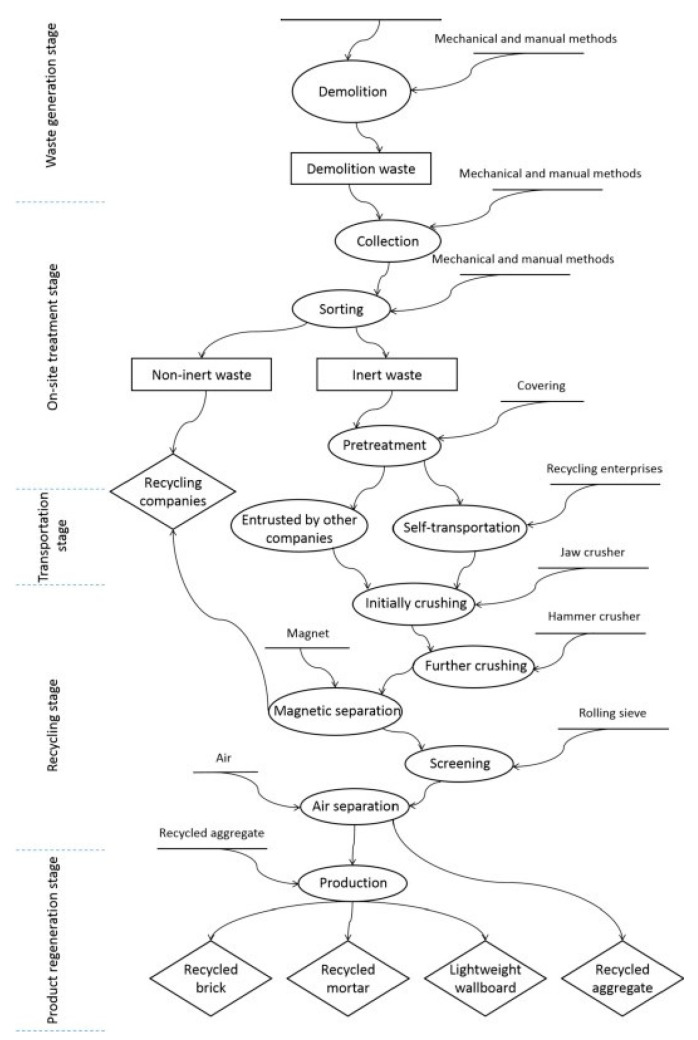
Offsite recycling [14].

**Figure 9 materials-13-02970-f009:**
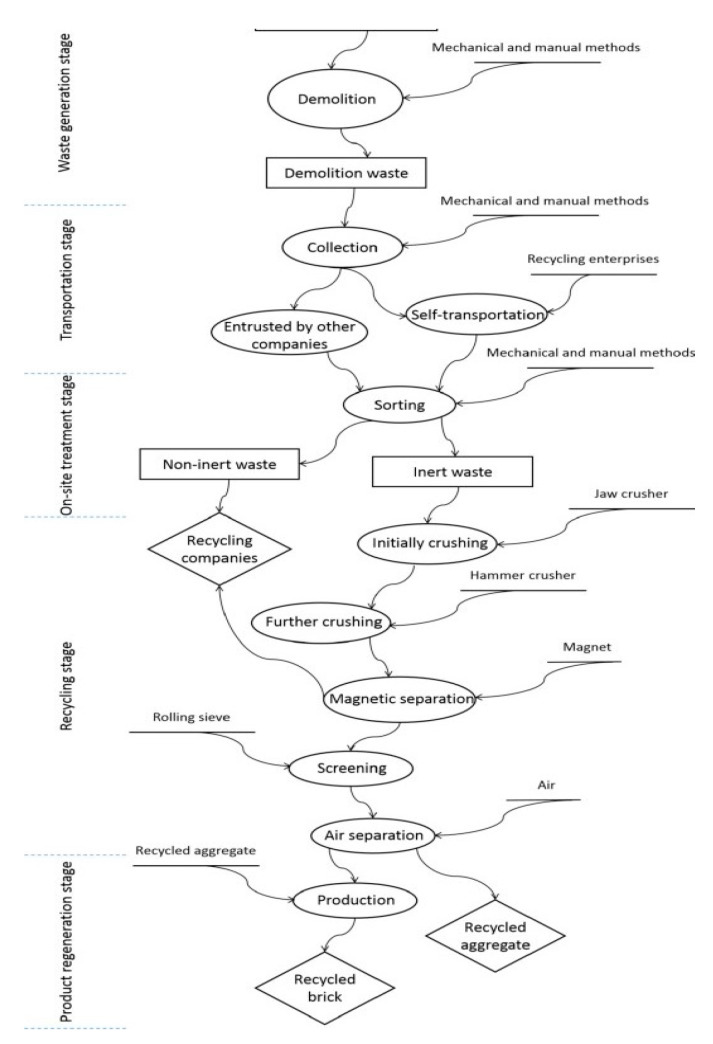
Onsite recycling [14].

**Figure 10 materials-13-02970-f010:**
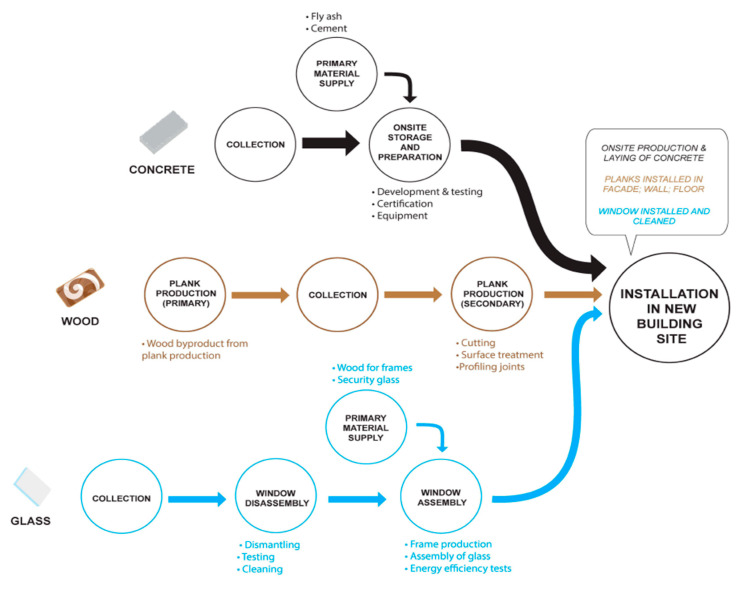
Single case study on concrete, wood, and glass [11].

**Table 1 materials-13-02970-t001:** Literature review of reuse and recycling CDW in new construction applications.

Reference	Location	Year	Reuse	Recycling	Type	Observations
[22]	India	2019		*	Brick dust and Concrete dust	Mechanical properties of resulting asphalt achieved equal or greater amounts.
[19]	U.S.	2019	*		Exterior wall frame systems	Reuse of exterior wall framing systems is feasible as long as transport distance is less than 3000 km.
[1]	Spain	2016		*	RCA	Physical, mechanical, and acoustic properties were the same as control material.
[25]	Poland	2016		*	Sewage sludge ash	Material can be used as substitute in various construction materials production.
[26]	Spain	2019		*	RA	Replacing 25 wt. % with recycled aggregate induces no significant effect.
[27]	Spain	2018		*	Ceramic aggregates	Material can be used in nonstructural functions.
[28]	Spain	2019		*	Polystyrene from C&D	Expanded polyterene and extruded polysterene waste from construction and demolition can substitute currently used aggregates perlite and vermiculite.
[29]	China	2020		*	RCA	Compressive strength of concrete blocks made with 100% RCA is within code requirements.
[20]	China	2019		*	RCA	Recycling of CDW is economically feasible.
[30]	Italy	2019		*	Concrete fibers	Recycled plastics and metal fibers in reinforced concrete can be used.
[24]	China	2019		*	RA	Current research focuses on adopting recycled aggregates in new concrete production.
[31]	Spain	2017		*	RA	In the production of nonstructural dry-mixed concrete hollow blocks, proper behavior was shown.
[23]	Spain	2016		*	CDW	Recycling CDW can be attractive when the recycled product is competitive with the virgin material in terms of cost and quality.
[32]	Italy	2018		*	RA	Using RCA into stabilized rammed earth material can be utilized.
[33]	Spain	2018		*	RA	Concrete workability was not affected by use of 100% mixed recycled aggregate. Neither compressive nor flexural strength varied significantly at replacement ratios.
[34]	Spain	2016		*	RA	Use of RA in production of lightweight mortars is a viable alternative.
[35]	Italy	2017		*	RCA	Production of self-compacting concrete with coarse and fine RCA up to 40% in volume.
[36]	Spain	2018		*	RCA	RCA from precast elements shows mechanical properties are slightly lower.
[37]	Iran	2019		*	RCA	Slight decrease in flexural strength can be observed.
[38]	France	2017		*	RA	16% additional cement was needed to compensate the drop in compressive strength of RAC.
[39]	Italy	2017		*	Fillers	Physical properties and mechanical performances are similar or even better compared to standard mixes.
[40]	Italy	2000		*	RA	Strict relationship between mix design is important and dosages correlate to the strength of the resulting material.
[41]	Spain	2017		*	RA	Substitution percentage below 35% show small decrease in mechanical properties.
[11]	Sweden	2019	*		Concrete, Wood, Glass	Material reuse has potential to become a price-competitive production practice.
[21]	U.K.	2016	*		Various construction and demolition materials	Reuse potential rates of commonly used construction materials has been established.
[42]	Poland	2018		*	Concrete rubble	By using a patented technology on thermal and mechanical treatment, waste-free recycling of rubble concrete into valuable materials such as RA and RCM were obtained.

Legend: RCA = Recycled concrete aggregates, RA = Recycled aggregates, RCM = Recycled cement mortar. * indicates which topic is covered by the referenced literature.

**Table 2 materials-13-02970-t002:** Summary of studies in the reuse and recycling of waste materials [43].

Location	Year	Type	RL	Observations
Finland	1998	RCA	100%	Self-cementitious properties of RCA led to an increasing load-bearing capacity.
Portugal	2009	RCA	100%	FWD test showed RCA layers have higher elastic moduli than the min. expected.
Singapore	2008	RCA	100%	The IRI sections was similar to or even better than the control sections.
Netherlands	1992	RMA	100%	Less strict requirements for alternative materials for mound dams are obtained due to successful experiences.
UK	2004	RCA/RAP	4%/29%	Savings of almost 2 million GBP.
Singapore	2008	RCA	100%	Similar IRI values were observed while the deflection was lower.
Spain	2009	MRA	100%	Higher water content for optimum compaction was required but appropriate mechanical performance and low deflections under impact was found.
Spain	2008	RCA	100%	Enhanced mechanical performance and similar to that expected of cement-treated gravel.
UK	2007	Tunnel spoil	100%	Decreased number of lorry movements by around 400 thousand and savings of about 10 million GBP
UK	1998	RCA	100%	FWD data showed the target elastic modulus of 2.5 GPa for combined bound layers.
UK	2004	RAP	35%	Use of alternative materials offered a savings of 0.5 GBP/sq. meter.
UK	2004	RAP	10%	Avoided the purchase of an equivalent to 54,000 tons of RAP and corresponding transportation cost.
UK	2003	RGA	100%	Indirect financial benefits included the removal of materials out of the waste stream.
Canada	2007	RCA	50%	Similar performance observed for all test sections suggesting no negative impact on the pavement’s performance.
USA	2016	RCA	40%	Compressive strength and shrinkage were higher and lower by 25% and 12% compared to control.
USA	1995	RCA	100%	Little difference in terms of performance
Austria	1991	RCA	100%	The subbase was 5% cheaper than the traditional alternative.
UK	2003	RCA	100%	The project showed that RCA concrete is suitable for use in an XF4 environment.
Spain	2014	MRA	100%	100% coarse MRA caused a decrease in strength but the difference between the two shortened in time.
Hong Kong	2006	RCA	100%	Using 100% with added 4% of cement resulted in comparable compressive strength.
Germany	1999	RCA/MRA	100%	Special decorative effects were achieved.
Germany	2000	MRA	30%	No significant differences found.
UK	1996	RCA	100%	100% coarse aggregate needed 10% cement content increase to achieve adequate performance.
UK	1996	MRA	20%	Similar strength development from conventional concrete was found.
Singapore	2010	RCA	100%	Yielded equivalent mechanical and durability performance to that of control concrete.
Denmark	1993	RCA	N/A	About 275 cubic meters of RAC made with coarse RA were used. Considerable experience was gained in the use of RAC.
Germany	2014	RCA	100%	The designed slump and target strength were achieved.
Germany	N/A	RCA	35%	About 500 cubic meters of concrete with 35% of coarse RCA were used.

Legend: RAP = Recycled asphalt pavement, MRA = Mixed Recycled Aggregate, RGA = Recycled glass aggregate.

**Table 3 materials-13-02970-t003:** Reuse potential rates of various construction components [21].

Reuse Potential Rates of a Range of Construction Components			
No Potential (0%)	Low (<50%)	Medium (=50%)	High (>50%)
Clay	Mineral Wool	Steel Cladding	Clay Bricks (lime-based mortar)
Steel rebar (buildings)	Gypsum wallboard	Steel cold form sections (buildings)	Structural timber
Steel rebar (other infrastructure)	Steel rebar in pre-cast concrete (buildings)	Steel pipes (buildings)	Structural steel (buildings)
Asphalt (other infrastructure)	Timber trusses	Slate tiles	Concrete building blocks (with lime mortar)
Asphalt roof shingles	Concrete in-situ	Timber floorboards	Concrete paving slabs and crash barriers
Plastic pipes (water and sewage), roof sheets, floor mats, electric-cable insulation, plastic windows	Concrete fencing, cladding, staircases, and stair units		Clay roof tiles
Concrete pipes and drainage, water treatment and storage tanks and sea and river defense units	Glass components (e.g., windows)		Stone paving
Nonferrous metal components (aluminum window frames, curtain walling, cladding, copper pipes, zinc sheets for roof cladding)			Stone Walling

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
