# Peer review of "Circular Economy on Construction and Demolition Waste: A Literature Review on Material Recovery and Production"

_materials, 2020, doi:10.3390/ma13132970_

Round 1

Reviewer 1 Report

The main issue about this paper is the demonstration of its innovation/added value, which is missing. As a matter of fact, most of the information in the paper (e.g. Figures 1, 2, 9, 10, 11 and 12, and Table 1) result from one single source (in other words, they are copied!) and the conclusions and so-called "research gaps and emerging topics" have been identified before in the literature. Unless this issue is properly addressed, the paper cannot be published.

The second main issue is the lack of explanation of the criteria used in the research methodology. Looking at Fig. 3, what were the criteria to select the first 360 articles? Then what were the criteria to chose 35 out of the 360? And what were the criteria to select the extra 7 older articles? The others say something about this in lines 83-93, but it is clear that there are MANY more than 35 articles that fulfill at least one of the four criteria mentioned.

Other issues that need to be addressed:

  • The English text needs to be thoroughly revised by an English-speaking native because it full of errors;
  • In Figure 3, the presentation of the topics should be deleted because it is meaningless to the readers;
  • In Figure 6, what is the meaning of the colours?
  • In Figure 7, why were these the only specific journals mentioned?
  • The quality of Figure 9 is quite bad;
  • What are the lessons learned from Table 1? The table is also poorly formatted;
  • Is it acceptable to copy a 2-page table (Table 2) that is not of the authors of this paper?
  • The formatting of the references is not correct.

Author Response

The main issue about this paper is the demonstration of its innovation/added value, which is missing. As a matter of fact, most of the information in the paper (e.g. Figures 1, 2, 9, 10, 11 and 12, and Table 1) result from one single source (in other words, they are copied!) and the conclusions and so-called "research gaps and emerging topics" have been identified before in the literature. Unless this issue is properly addressed, the paper cannot be published.

Response to the comment: The purpose of this review is to study some of the frameworks adopted in CE and CDW specifically in the reuse and recycling of materials. The contribution of our work can be seen in the conclusion where we highlighted the findings that we were able to discover from reading and analyzing the latest research done in the field (e.g. feasibility of recycled materials up to 40% substitution, additional materials to compensate slight decrease in mechanical properties, etc.). Our review can guide future research and experimentation into focusing on improving the substitution rate of recycled materials thus reducing or eliminating the need for additional materials to be added to compensate for the decrease in mechanical properties and the like as discussed in our conclusion. Figures 1, 9, 10, 11, and 12 were from different authors who we believed provided effective frameworks which we can learn from in future research works. Table 1 is did not came from a single source since it is a summary of the different experiments made in reusing and recycling materials from the references that we have.

The second main issue is the lack of explanation of the criteria used in the research methodology. Looking at Fig. 3, what were the criteria to select the first 360 articles? Then what were the criteria to chose 35 out of the 360? And what were the criteria to select the extra 7 older articles? The others say something about this in lines 83-93, but it is clear that there are MANY more than 35 articles that fulfill at least one of the four criteria mentioned.

Response to the comment: At the time the review was made, using the keywords in Scopus "circular economy AND construction OR demolition AND waste OR recycling", around 400+ documents will be shown. Limiting the search from documents published from 2016 to 2020, 360 documents were found. This was shown in fig. 3. In choosing the 35 articles, the criteria in lines 67-73 were used and we considered what we thought were the most relatable articles to our chosen topic, other articles were a little off-topic or emphasizes other topics than focusing on circular economy and construction and demolition waste. the 7 older articles included were also considered using the criteria in lines 67-73 but we gave emphasis on what we saw were the earliest published articles on CE, CDW, and reuse and recycling of CDW in new construction applications.

The English text needs to be thoroughly revised by an English-speaking native because it full of errors;

Response to the comment: Edited the manuscript and improved English. However, If language errors still become an issue, we are willing to avail MDPI English editing service to improve the paper.

In Figure 3, the presentation of the topics should be deleted because it is meaningless to the readers

Response to the comment: fig. 3 is removed and explanation on how we came up with the chosen articles are explained in lines 58-81.

In Figure 6, what is the meaning of the colours?

Response to the comment: the colors represents the topics as discussed in lines 92-95 and a legend is added in fig. 6 lines 153-156 for easier understanding.

In Figure 7, why were these the only specific journals mentioned?

Response to the comment: Fig. 7 was removed because we think that this is irrelevant and the number of published papers per year is the only relevant data to distinguish the trend of the topic if it is an emerging topic.

The quality of Figure 9 is quite bad

Response to the comment: changed the figure with a high-resolution version.

What are the lessons learned from Table 1? The table is also poorly formatted

Response to the comment: The table has been formatted properly. Lessons learned from table 1 as well as from the research done by Silva et al., 2019 were in lines 285-353. 

Is it acceptable to copy a 2-page table (Table 2) that is not of the authors of this paper?

Response to the comment: Rectified the issue by removing one of the figures retaining only the figure that most represents the greater challenge in CDW management.

Reviewer 2 Report

The article could be interesting for the journal, however the authors have to take into account some aspects to improve it:

On the one hand, and given that the authors consider the circular economy as the subject to which construction sector must progress, in my opinion, this concept should be included in the title.

On the other hand, the authors have to take into account that the article interest is not in the methodology they have followed to find scientific studies about CE, etc, but in the studies themselves and in their conclusions, taking into account them, in order to propose future lines of research.

Therefore, in my opinion, the methodology chapter should be reviewed and specified. For example, Figures 4, 5, 6, 7 and 8 should be removed or at least reduced, since they contribute very little to the objective of the study.

I also think that chapters 3 and 4 should be improved, and the authors must summarize the results of the papers found, incorporating the most important issues in order to conclude on interesting topics for the building sector.

I would also like the authors to reconsider the statement "practical application of reuse in studies are scarce as compared to the numerous research and experimentation done on recycling CDW", in my opinion to affirm this thesis, the authors should take into account, not only the work of Iacovidou & Purnell (2016), a study that summarizes the different reuse potential rates or embodied carbon (EC) reuse efficiency of a range of construction components and that only considers studies from 2000 to 2014, I think there are many other studies before and after 2014, which consider the study of practical applications. So this section should be expanded and at the end, authors could conclude about the issue.

Also the authors must reconsider the following statement in the conclusions chapter... Recycling materials leads to mostly the same physical and mechanical properties as those of their virgin counterparts, because it is not entirely true, despite the fact that they have added .......In cases where the mechanical properties are slightly lower than their natural counterparts, a small amount of other materials are used to compensate for the slight decrease which is almost negligible compared to the environmental benefits of recycling CDW. In my opinion, the topic is very complicated and diverse, and the authors cannot be summarized it so categorically.

Other questions to consider:
They should write what CE means, before using the acronym.
Review the references in the text.

Author Response

On the one hand, and given that the authors consider the circular economy as the subject to which construction sector must progress, in my opinion, this concept should be included in the title.

Response to the comment: To address this comment we may be able to adjust the title to "Circular economy on construction and demolition waste: A literature review on material recovery and production towards progress in the construction sector" so that we may give focus to the contributions that the circular economy can provide towards the progress of the construction sector.

On the other hand, the authors have to take into account that the article interest is not in the methodology they have followed to find scientific studies about CE, etc, but in the studies themselves and in their conclusions, taking into account them, in order to propose future lines of research.

Response to the comment: The methodology part in our literature review discusses on how we gathered or acquired our references. These references are evaluated based on at least qualifying to one (1) of the criteria in lines 67-73. Aside from the criteria, we have manually assessed the papers to ensure that there are no redundant references and to ensure that the references focus on circular economy and on reuse and recycling of CDW in new construction applications. Based on the studies, experiments, and references that we have, we were able to come up or propose future work for us and for other researchers.

Therefore, in my opinion, the methodology chapter should be reviewed and specified. For example, Figures 4, 5, 6, 7 and 8 should be removed or at least reduced, since they contribute very little to the objective of the study.

Response to the comment: Fig. 5 removed. The purpose of having fig. 4 is to show how we arrived in lines 92-95 which shows the four primary topics among the documents that we added as reference to the review. fig. 3 shows the probability of each primary topic to appear per document in the included references. Fig. 7 is removed and 8 (changed to fig.4)  shows the trend of published documents just to show that research in CE increases every year.

I also think that chapters 3 and 4 should be improved, and the authors must summarize the results of the papers found, incorporating the most important issues in order to conclude on interesting topics for the building sector.

Response to the comment: Chapters 3 and 4 improved by additional inputs from references. We have provided a summary as seen in table 1 where we summarized various reuse and recycling research and experimentation. With the references that we have we were able to arrive in a conclusion as seen in chapters 5 and 6. With those chapters we are able to impart a guide to future studies and research.

I would also like the authors to reconsider the statement "practical application of reuse in studies are scarce as compared to the numerous research and experimentation done on recycling CDW", in my opinion to affirm this thesis, the authors should take into account, not only the work of Iacovidou & Purnell (2016), a study that summarizes the different reuse potential rates or embodied carbon (EC) reuse efficiency of a range of construction components and that only considers studies from 2000 to 2014, I think there are many other studies before and after 2014, which consider the study of practical applications. So this section should be expanded and at the end, authors could conclude about the issue.

Response to the comment: Rephrased "Practical application of reuse in studies are scarce as compared to the numerous research and experimentation done on recycling CDW". We may have used the wrong word or words thus giving a misleading statement. Instead we have rephrased the lines. We agree that there are many studies in reuse but majority of the documents we found and assessed based on the methodology that we used in chapter 2 were focused on recycling CDW in new construction applications.

Also the authors must reconsider the following statement in the conclusions chapter... Recycling materials leads to mostly the same physical and mechanical properties as those of their virgin counterparts, because it is not entirely true, despite the fact that they have added .......In cases where the mechanical properties are slightly lower than their natural counterparts, a small amount of other materials are used to compensate for the slight decrease which is almost negligible compared to the environmental benefits of recycling CDW. In my opinion, the topic is very complicated and diverse, and the authors cannot be summarized it so categorically.

Response to the comment: The conclusion is based on the different studies and experiments that we summarized in Table 1 as well as past compilation of different research and experiments made by Silva et al., 2019 where there are mostly 3 outcomes to the studies:

  • Recycled Materials leads to almost the same physical and mechanical property. (Which should be studied further because there are instances that mechanical properties are lower.)
  • Replacement of 40% or less recycled materials leads to acceptable mechanical properties when tested.
  • Additional materials (e.g. more cement) to compensate for the decrease in strength or other mechanical property.

all of which should be studied further as mentioned in chapter 5 since there are different and varying results from different researchers.

They should write what CE means, before using the acronym.
Review the references in the text.

Response to the comment: In the abstract instead of writing "CE" prior to its meaning, it is changed to “Circular Economy". We have dedicated lines 34-54 to properly describe and to give a brief history of CE as well as to show the rising awareness and popularity of the system.

References were reviewed and revised to follow standard citation and inclusion in the reference list.

Reviewer 3 Report

The article is chaotic. It contains numerous language errors. The text also contains many editing errors - various fonts, whole tables pasted from other articles (e.g. Fig 12- not acceptable) - which should also be a table and not a drawing. The quality of graphs is not acceptable. References list also is not according to requirements.

There are a lot of Authors which writing that amount of CDW is more than 530 millions tones.

In which country is 530 million tones- it seems per year?

The literature review of reuse CRA is not complete and not actuall (Tab.1.)

There are a lot of new methods of recycling concrete aggregates. Authors do not open this topic- but it is very important.

Author Response

The article is chaotic. It contains numerous language errors. The text also contains many editing errors - various fonts, whole tables pasted from other articles (e.g. Fig 12- not acceptable) - which should also be a table and not a drawing. The quality of graphs is not acceptable. References list also is not according to requirements.

Response to comment: fonts, tables, and language errors corrected. fig. 12 removed and converted to a table. Reference list formatted according to standards. If language errors still become an issue, we are willing to avail MDPI English editing service to improve the paper.

There are a lot of Authors which writing that amount of CDW is more than 530 millions tones.

Response to comment: re-phrased the sentence and provided additional data from the references.

In which country is 530 million tones- it seems per year?

Response to comment: Already addressed this comment in the previous comment. 930 million tons in EU in 2016 and added 2.36 billion tons in China in 2018.

The literature review of reuse CRA is not complete and not actuall (Tab.1.)

Response to comment: Table 1 formatted. The summarized items in table 1 came from all references that studied or experimented in reusing or recycling CDW. References which only discussed CE or discussed frameworks in CE or CDW were not included in the table.

There are a lot of new methods of recycling concrete aggregates. Authors do not open this topic- but it is very important.

Response to comment: Added inputs in the paper to discuss methods in recycling concrete aggregates and CDW.

Round 2

Reviewer 1 Report

The authors were unable or unwilling to respond satisfactorily to the following issues, raised in the previous review: Demonstration of the innovation; criteria used to select the references; quality of the English text.

Therefore, the paper must be rejected.

Author Response

The authors were unable or unwilling to respond satisfactorily to the following issues, raised in the previous review: Demonstration of the innovation; criteria used to select the references; quality of the English text.

Therefore, the paper must be rejected.

Response to the comment: 

1.) Demonstration of the innovation: The innovation/added value in our paper can be seen in the following:

a.) We used Matlab text data analytics in determining the underlying topics of the references chosen in this review article. As seen in fig. 3, we were able to produce topic mixtures including probabilities of underlying topics. This method can be used in future studies for ease of identification of underlying topics in a given set of references especially when the number of references/papers selected are many.

b.) We summarized various studies and experimentation recently done (from 2016 to 2020). This summary can be used as guide to future studies on what can be improved in the field of CE, CDW, Reuse and Recycling of CDW. Having a summary of various studies in the span of the recent years can be very advantageous in reducing if not eliminate redundancy of studies/experimentation. This can also be used to produce improved frameworks and methods in reuse and recycling since authors now have an idea of the innovations or improvements made by different authors in the past years. Thus, paving a way to improved future studies and avoiding errors made from past ones.

c.) In the part 5 and 6, we have discussed the emerging topics in the field as well as research gaps that can be undertaken by future researchers and future work. Expert suggestions on future work based on the data we gathered from the included references were discussed so as to provide detailed guide to future research and experimentation.

2.) Criteria used to select the references: The criteria we used in selecting the references were thoroughly discussed in 2.Methodology. Lines 58 to 64 discussed the process from the use of keywords in Scopus to narrowing down the papers from 2016 to 2020 only. Lines 62 to 79 discussed in detail the criteria we considered in coming up with the references that were included in our review article.

3.) Quality of English text: We considered the quality of English in our paper and edited applying correct grammar, punctuation, and style to our review article.

Reviewer 2 Report

In my opinión the article has improved.

Author Response

Thank you for the comments and suggestions that greatly improved our paper.

Reviewer 3 Report

The Authors improved the paper.

I suggest to put into Table 1, a new modern method of recycling of concrete which was made in Poland in 2018  and is described in :                 a) Kalinowska-Wichrowska, K.;Pawluczuk, E.;BoÅ‚tryk,M. Waste -free technology for recycling concrete rubble, Build. Mater.2020, doi.org/10.1016/j.conbuildmat.2019.117407

Author Response

I suggest to put into Table 1, a new modern method of recycling of concrete which was made in Poland in 2018  and is described in : a) Kalinowska-Wichrowska, K.;Pawluczuk, E.;BoÅ‚tryk,M. Waste -free technology for recycling concrete rubble, Build. Mater.2020, doi.org/10.1016/j.conbuildmat.2019.117407

Response to the comment: Suggested paper is analyzed, studied, and added in table 1 and as reference. The suggested paper is also highlighted in lines 263 to 268 due to its innovation in waste-free recycling. 

Thank you for the comments and suggestions in previous round that improved our paper.